# Intracellular Enzyme-Instructed Self-Assembly of Peptides (IEISAP) for Biomedical Applications

**DOI:** 10.3390/molecules27196557

**Published:** 2022-10-04

**Authors:** Fengming Lin, Chenyang Jia, Fu-Gen Wu

**Affiliations:** State Key Laboratory of Bioelectronics, School of Biological Science and Medical Engineering, Southeast University, 2 Sipailou Road, Nanjing 210096, China

**Keywords:** cancer theranostics, bioimaging, antibacterial, hydrogel, nanostructure, peptide

## Abstract

Despite the remarkable significance and encouraging breakthroughs of intracellular enzyme-instructed self-assembly of peptides (IEISAP) in disease diagnosis and treatment, a comprehensive review that focuses on this topic is still desirable. In this article, we carefully review the advances in the applications of IEISAP, including the development of various bioimaging techniques, such as fluorescence imaging, photoacoustic imaging, magnetic resonance imaging, positron-emission tomography imaging, radiation imaging, and multimodal imaging, which are successfully leveraged in visualizing cancer tissues and cells, bacteria, and enzyme activity. We also summarize the utilization of IEISAP in disease treatments, including anticancer, antibacterial, and antiinflammation applications, among others. We present the design, action modes, structures, properties, functions, and performance of IEISAP materials, such as nanofibers, nanoparticles, nanoaggregates, and hydrogels. Finally, we conclude with an outlook towards future developments of IEISAP materials for biomedical applications. It is believed that this review may foster the future development of IEISAP with better performance in the biomedical field.

## 1. Introduction

Molecular self-assembly is a bottom-up, controllable, and effective way to produce functional ordered supramolecular materials with various dimensions from nanoscale to microscale, in which molecules spontaneously and hierarchically assemble by both internal and external interactions [1,2]. In nature, different biomolecules, such as biopolymers, peptides, proteins, and nucleic acids (DNA and RNA), can self-assemble into diverse nanostructures, inspiring the design, synthesis, and applications of self-assembled biomaterials in the fields of nanotechnology, environmental science, energy storage, biomedicine, and others [3,4,5,6]. Different stimuli can be harnessed to trigger molecular self-assembly in vitro or in vivo for obtaining desired materials [7,8,9,10], such as the building block concentration [11,12], enzyme [13], solvent [14], temperature [15], pH [16,17], ionic strength [17], metal ion [7], physical stimulation [8], and ligand–receptor interaction [9]. Recently, many research efforts have been devoted by a variety of researchers to the structural design, functional tailoring, and applications of enzyme-instructed self-assembly (EISA) of peptides, due to its following merits [10,13]: First, the structure of peptides can be modified easily by changing amino acid sequence. Second, the self-assembly of peptides into nanofibers, nanoparticles, nanoaggreates, and hydrogels can be tuned by controlling experimental conditions. Third, the peptide-derived materials have satisfactory bioactivity and biocompatibility.

Peptides consist of several or dozens of amino acids linked by amide bonds. They are ubiquitous in organisms and have critical biological functions. In addition, they are inherently bioactive, biocompatible, and biodegradable, making them ideal building units for functional biomaterials. There are twenty natural amino acids for peptide synthesis, which share the same backbone structure but vary in the R groups (side groups), enabling the construction of an enormous number of peptide sequences with diverse properties, suitable for use as assembling blocks. With specific design, peptides can have the ability to mimic the self-assembly behavior of proteins, further making them excellent choices for constructing materials with highly ordered structures and diverse functions. For instance, aromatic dipeptides [18], peptide-amphiphiles [19], and polypeptides [15] have been demonstrated to be molecular building blocks with excellent self-assembly properties, which can produce nanoscale structures, such as nanofibers and nanoparticles, through various noncovalent interactions between amino acid residues, such as ionic, hydrogen bonding, hydrophobic, and π–π stacking interactions. In addition to the self-assembling peptide units, self-assembled peptide-based precursors generally also consist of functional components and responsive/targeting groups. Peptides are easy to manipulate and synthetically accessible due to their simple linear structures. They can be either isolated from living organisms or synthesized through chemical methods, such as solid-phase peptide synthesis, ring-opening polymerization, and solution phase synthesis [20,21].

Enzymes have been considered as one of the most versatile strategies to control peptide self-assembly due to their reliable selectivity and mild reaction conditions in the catalysis of chemical reactions, as well as their good biocompatibility and spatiotemporality in cellular environments [10]. Since the expression and distribution of enzymes differ by the types and states of cells, tissues, and organs, when using an enzymatic reaction to realize the self-assembly of peptides into hierarchical nanamaterials, one can manipulate the delivery, function, and response of a nanomaterial according to a specific biological environment, therefore offering an accessible route to create desired materials for biomedical uses. In particular, intracellular enzymatic self-assembly of peptides provides a unique means for researchers to combine molecular self-assembly with intrinsic enzymatic reactions inside cells for developing novel biomaterials at the supramolecular level. In this context, different enzymes have been leveraged in manipulating peptide self-assembly in living cells, including alkaline phosphatase (ALP), caspase, furin, hyaluronidase (HAase), among others [10,22,23]. For instance, the most frequently used enzyme for intracellular enzyme-instructed self-assembly of peptides (IEISAP) is ALP. ALP is overexpressed on the cell membranes of tumor cells, such as Saos-2, HeLa, Hep G2, and MESSA/Dx5 [24,25,26,27], bacteria [28,29,30], and tear fluid [31]. ALP dephosphorylates the peptide-derived precursors, generating more hydrophobic products for peptide assembly/aggregation and realizing in situ self-assembly of peptides into supramolecular nanostructures in living systems.

Although traditional strategies that employ nanostructures assembled ex situ have demonstrated increased bioavailability and targetability due to the enhanced permeability and retention effect and the multivalent effect, the preassembled structures obtained from ex situ assembly sometimes suffer from inherent instability under sophisticated physiological environments in vivo. Exploiting the dynamic nature of molecular self-assembly, in situ assembly/reassembly has shown promising results in the prolonged retention of imaging and therapeutic agents, thus promoting its potential applications in long-term imaging and sustained therapeutic release. The main advantage of the IEISAP strategy is that the assembly can be precisely controlled to occur in the vicinity of a specific physiological or even pathological site by the fine tuning of peptide building units with targeted accumulation and responsive retention properties, leading to a low detection limit, high imaging quality, or great therapeutic efficacy [32]. The general processes for in vivo enzyme-activated self-assembly of peptides are described as follows: the peptide-derived precursors often contain a substrate motif, which will be recognized by a specific enzyme of interest. Upon arriving at the target cells, the precursors are converted by cellular enzymes into amphiphilic building blocks that self-assemble spontaneously into functional ordered supramolecular materials by noncovalent interactions (hydrophobic interaction, π−π interaction, hydrogen bonding, and electrostatic interaction), which can endow the peptides with improved stability, increased mechanical strength, or enhanced activity. In addition to noncovalent interactions, biocompatible condensation [33] and intracellular macrocyclization reaction [34] strategies have also been established to realize the desirable structure formation in living systems.

Since Bing Xu’s group first demonstrated the enzyme-instructed self-assembly of aromatic peptide amphiphiles within biological systems in 2007 [35], IEISAP has found many applications, due to the development of various peptides/peptide derivatives and the discovery of various enzymes that can be used for IEISAP. IEISAP allows for the creation of peptide nanostructures with diverse functions for high-performance biological applications, ranging from disease diagnosis to treatment, such as cancer tissue/cell imaging, enzyme activity assay, cancer therapy, and antibacterial application, and it has high potential in reducing drug toxicity, improving drug targeting, and enhancing drug delivery efficiency. However, a comprehensive review that specifically focuses on the use of IEISAP systems for biomedical applications is still lacking, although several reviews regarding self-assembled peptide-based materials (which do not specifically focus on the materials prepared via IEISAP) for biomedical applications have already been published [13,20,21,36,37,38]. In this review, we will specifically focus on the developments of in situ enzyme-activated self-assembly of peptides for biomedical applications, including the imaging and treatment of diseases (mainly cancers) (Figure 1). We will illustrate the design, working mechanisms, and functions/applications of IEISAP. Finally, we will propose some current challenges and future research directions in this field.

## 2. IEISAP for Imaging Applications

In terms of biomedical applications, bioimaging technologies are urgently calling for highly efficient probes/contrast agents for high-performance bioimaging, including molecular imaging, cell imaging, and tissue/organ imaging. IEISAP has been employed in improving the performance of routine modalities for bioimaging, such as fluorescence imaging, photoacoustic (PA) imaging, magnetic resonance imaging (MRI), positron-emission tomography (PET) imaging, computed tomography (CT) imaging, and multimodal imaging, as we will review below.

### 2.1. Fluorescence Imaging

The fluorescent IEISAP materials display varied structures and functionalities, and can be employed as potential fluorescent probes for high-performance biomedical imaging with prolonged tumor retention [39], high (photo)stability [40,41,42,43], and targeted biological distribution [44,45]. Generally, the building blocks for fluorescent IEISAP materials are peptide−fluorophore conjugates. The peptide units for self-assembly can be covalently linked with different fluorogens, such as 4-nitro-2,1,3-benzoxadiazole (NBD) [44,46,47,48,49,50], aggregation-induced emission luminogens (AIEgens) [45,51,52,53,54,55,56], the near-infrared (NIR) dye cyanine (Cy) [40,41,42], the fluorescent dye Alex Fluor 647 [39], the coumarin dye [57], 1,8-naphthalic anhydride [58], and fluorescein isothiocyanate (FITC) [43], to generate peptide−fluorophore conjugates as the building blocks for IEISAP. Specific enzymes in vivo and/or in cells can trigger the self-assembly of these peptide−fluorophore conjugates into nanostructures such as nanofibers [41,43,44,46,47,48,50,55,56,57,58] and nanoparticles [42,53] in which the fluorescence can be enhanced/decreased. These enzymes include ALP [43,46,50,51,55,57,58,59,60], caspase-3/7 [40,42,52,53], cathepsin B [45,54], matrix metalloproteinases (MMPs) [39,41], esterase [47], enterokinase (ENTK) [48], sirtuin family, which consists of seven isoform 5 (SIRT5) [44], and autophagy-related 4 homolog B (ATG4B) [56]. IEISAP has been utilized to monitor enzyme activity [42,43,44,50,55,57], self-assembly of peptides [46,60], subcellular compartments [44,48], apoptosis [52,53], and autophagy [56] in living cells, as well as visualize tumor tissues and cells [39,40,41,43,45,51,53,54].

For instance, Yang et al. reported SIRT5-mediated self-assembly of fluorescent peptide precursors for imaging the SIRT5 activity in mitochondria [44]. SIRT5, a mitochondria-oriented enzyme, belongs to a family of nicotinamide adenine dinucleotide (NAD^+^)-dependent histone deacetylases, and is closely associated with the regulation of diverse biological processes, such as apoptosis, fatty acid metabolism, and reactive oxygen defense. Nevertheless, designing biosensors for detecting intracellular SIRT5 activity is challenging, yet lacking. The peptide precursor consisted of an environment-sensitive fluorophore NBD for imaging, a phenylalanine-rich peptide fragment, and a Ksucc (succinylated lysine) switch module (Figure 1a). These amphipathic peptide precursors with low molecular weight could effectively enter cells, in which their aggregation in the cellular environment enhanced their cell internalization. Upon arriving at mitochondria, the negatively charged Ksucc in the peptide precursor was desuccinylated by SIRT5 to generate the positively charged lysine residue, forming a desuccinylated peptide building block with zero charge (Figure 1a). The unique zwitterionic nature of the peptide building block increased the electrostatic interaction between each other, leading to the self assembly into nanofibers. Meanwhile, the NBD in the nanofibers produced bright fluorescence due to the hydrophobic environment in nanofibers, realizing fluorescence detection or imaging of cellular SIRT5 activity (Figure 1b). In addition to bioimaging applications, fluorescent IEISAP materials have also been applied in cancer theranostics [40,45,51,53,54], as reviewed in Section 3.1.4.

### 2.2. Photoacoustic (PA) Imaging

Photoacoustic (PA) imaging is a noninvasive imaging method by detecting ultrasonic waves produced from the transient thermoelastic expansion of biological tissues upon light absorption from a pulsed laser. PA imaging simultaneously possesses the sensitivity of fluorescence imaging, and the high spatial resolution (up to tens of micrometers) and deep tissue penetration (up to a few centimeters) of ultrasound imaging, holding great promise for the accurate evaluation of important physiological and pathological processes. Nevertheless, there are only a few naturally occurring light absorbers, such as hemoglobin and melanin. Thus, various exogenous photoacoustic agents, based on materials including IEISAP assemblies, have been developed to enhance the photoacoustic contrast. The photoacoustic agents based on IEISAP materials that can respond to various enzymes, such as gelatinase [61,62], autophagy-specific enzyme ATG4B [63], ALP [64], caspase-1 [65], caspase-3 [66], and furin [67], have been utilized for imaging tumors [62,63,64,66,67] and detecting bacteria [32,65]. In addition to enhanced photoacoustic signals, the IEISAP strategy endows photoacoustic agents with high stability in vivo [32,67], desirable targeting properties [65,67], and long retention in tumor sties [62,66]. Wang et al. developed a caspase-3-activatable PA imaging probe (termed 1-RGD) for real-time and high-resolution imaging of tumor apoptosis. 1-RGD contained a tumor-targeting cyclic peptide (cyclic Arg-Gly-Asp, c-RGD), 2-cyano-6-hydroxyquinoline (CHQ), a D-cysteine (D-Cys) residue, a caspase-3-cleavable peptide substrate (Asp-Glu-Val-Asp, DEVD), a glutathione (GSH)-reducible disulfide bond, and a clinically used NIR dye (indocyanine green, ICG) (Figure 2a) [66]. In apoptotic tumors, intracellular GSH and active caspase-3 uncaged the thiol and amino groups of the D-Cys residue in 1-RGD, respectively (Figure 2b). The free d-Cys could interact with CHQ by fast intramolecular condensation to produce a cyclized product 1-cycl that was more hydrophobic and rigid than 1-RGD, causing stronger intermolecular interactions, such as hydrophobic interaction and π–π stacking, to enhance the molecular self-assembly into nanoparticles. Compared with 1-RGD, the density of ICG molecules in the nanoparticles was higher, leading to the lower NIR fluorescence because of the aggregation-caused quenching (ACQ) effect and increased PA signals, due to the augmented nonradiative relaxation processes (Figure 2c). Meanwhile, the larger size of the nanoparticles, compared with that of 1-RGD, endowed them with prolonged retention in apoptotic tumor regions. Collectively, confined PA signal improvement in apoptotic tumor tissues was realized to monitor caspase-3 activity for evaluating the apoptosis status in the whole tumor tissue, facilitating early and real-time evaluation of tumor therapeutic efficacy, prior to the alteration in tumor size.

Besides tumor diagnosis, PA imaging based on IEISAP has been employed for bacterial infection diagnosis [32,65]. Currently, the available effective approaches for early-stage pathogen diagnosis cannot be applied in vivo and suffer from low sensitivity and specificity [32]. Therefore, new strategies are urgently needed to be developed for accurately visualizing bacterial infections in situ. Li et al. developed a photoacoustic contrast agent, assembled from an enzyme-responsive peptide, for in vivo specific and sensitive imaging of bacterial infection [32]. The building block (Ppa-PLGVRG-Van) contained pyropheophorbide-α (Ppa) as a signal ligand, Pro-Leu-Gly-Val-Arg-Gly (PLGVRG) as an enzyme-activatable peptide linker, and vancomycin (Van) as a targeting molecule. Ppa-PLGVRG-Van could be selectively anchored to the Gram-positive bacterial cell wall through multiple hydrogen bonding interactions between Van and the terminal D-alanyl-D-alanine moieties of the *N*-acetyl muramic acid (NAM)/*N*-acetyl-glucosamine (NAG) peptides on the cell wall of Gram-positive bacteria. Then, the PLGVRG linker was specifically cleaved by gelatinase in gelatinase-overexpressing bacteria to produce Ppa-PLG. With enhanced hydrophobicity and decreased steric hindrance, Ppa-PLG self-assembled into twisted fibers, resulting in an increased heat conversion efficiency and an enhanced photoacoustic signal. Accordingly, Ppa-PLGVRG-Van could successfully image bacterial infections with high sensitivity and specificity. Moreover, this contrast agent could discriminate bacterial infection from sterile inflammation in vivo, and could also detect Gram-positive, gelatinase-expressing bacteria with high sensitivity. The PA contrast agents based on IEISAP offer new possibilities for the specific and sensitive diagnosis of bacterial infections in vivo.

### 2.3. Magnetic Resonance Imaging (MRI)

MRI is a widely used, powerful, and noninvasive imaging method for clinical diagnosis of tumors with unlimited penetration depth and high spatial resolution. Nevertheless, the sensitivity of MRI is low, requiring constant utilization of contrast agents (CAs) to promote the imaging quality and accuracy by altering the spin–lattice relaxation time (*T*1) or spin–spin relaxation time (*T*2). To solve this issue, IEISAP has been harnessed to increase the relaxation efficiency of traditional *T*1 CAs, such as Gd(III)-based CAs [68,69,70], and *T*2 CAs, such as superparamagnetic iron oxide (SPIO) nanoparticle-based CAs [71,72], for tumor imaging with the help of enzymes that are overexpressed in cancer cells, such as ALP [68,73], caspase 3/7 (Casp3/7) [70,71], MMP-2 [69], and furin [72]. For example, Ding et al. functionalized Fe_3_O_4_ nanoparticles (IONPs) with a dual-functional fluorine probe 4-(trifluoromethyl)benzoic acid (TFMB)-Arg-Val-Arg-Arg-Cys(StBu)-Lys-CBT to obtain IONP@1 (Figure 3) [72]. The as-synthesized IONP@1 was composed of the TFMB-Arg-Val-Arg-Arg (TFMB-RVRR) substrate for furin cleavage and providing ^19^F nuclear magnetic resonance (NMR)/MRI signal, a 2-cyanobenzothiazole (CBT) residue linked with a caged cysteine moiety for click condensation reaction, and Fe_3_O_4_ NP (IONP) as a *T*2 MRI CA. When IONP@1 encountered furin-overexpressing cells, the intracellular GSH reduction of the disulfide bonds in IONP@1 and furin-mediated cleavage of TFMB-RVRR resulted in the generation of 1,2-aminothiol groups. The crosslinking of IONPs was achieved by the click condensation reaction between the 1,2-aminothiol groups and cyano moieties, forming IONP aggregates. The formation of IONP aggregates caused a lower *T*2 value of the surrounding water protons, and consequently a stronger *T*2 magnetic resonance signal. Meanwhile, the peeling-off of TFMB-RVRR from IONP relieved the paramagnetic relaxation enhancement effect, thereby turning “on” the ^19^F NMR/MRI signal. In this way, IONP@1 was successfully employed for precise dual-mode (^1^H and ^19^F) MRI of tumors in zebrafish under 14.1 T. This work proposes a solution to address the dilemma between selectivity and sensitivity of traditional MRI sensors.

### 2.4. Positron-Emission Tomography (PET) Imaging

PET is a routine method of tumor imaging in clinic, offering valuable information to discriminate changes in cancer at the cellular level by utilizing radiotracers to image biological processes with ultrahigh sensitivity. However, few tumor-targeted PET imaging probes are available for precisely visualizing a specific tumor, urgently requiring the development of tumor-targeted PET imaging probes to increase the specificity of PET imaging. To this end, IEISAP has been implemented to develop tumor-targeting radioactive probes for enhanced microPET imaging of tumors [74,75]. As an example, Wang et al. rationally developed a furin-responsive radiotracer Acetyl-Arg-Val-Arg-Arg-Cys(StBu)-Lys-(DOTA-^68^Ga)-CBT (CBT-^68^Ga; DOTA is the abbreviation of “1,4,7,10-tetraazacyclododecane-1,4,7,10-tetraacetic acid”) and coinjected it with its cold analogue CBT-Ga in furin-overexpressing MDA-MB-468 cancer cells, leading to the formation of ^68^Ga nanoparticles (CBT-^68^Ga-NPs) [75]. The formation of CBT-^68^Ga-NPs notably improved microPET imaging performance of the tumor in vivo. In brief, in cancer cells overexpressing furin, disulfide bond reduction and furin cleavage of Arg-Val-Arg-Arg (RVRR) occurred, converting CBT-Ga to the active intermediate Cys-Lys(DOTA-Ga)-CBT. Cys-Lys(DOTA-Ga)-CBT immediately underwent a CBT–Cys condensation reaction to generate the cyclized oligomers (CBT-Ga-Dimer and CBT-Ga-Trimer) that self-assembled into nanoparticles (CBT-Ga-NPs) with an average diameter of 258.3 nm under the physiological condition. In vivo microPET imaging data showed that the mice coadministrated with CBT-^68^Ga and CBT-Ga had a tumor/liver ratio 9.1-fold of that of the mice administrated with CBT-^68^Ga, realizing improved tumor microPET imaging. The presence of the cold analogue CBT-Ga was needed for the intracellular formation of CBT-^68^Ga-NPs by overcoming the interference of intracellular Cys with the CBT–Cys condensation of the low-concentration CBT-^68^Ga in vivo. This work successfully applies IEISAP to improve the microPET imaging performance. It is also expected that IEISAP will offer a strategy to devise a variety of radioactive agents for more sensitive and precise microPET imaging of tumors in the future.

### 2.5. Computed Tomography (CT)

X-ray computed tomography (CT) images body structures and tissues by utilizing the different absorption effects from different human tissues with the assistance of contrast agents. As a convenient and efficient diagnostic method, CT imaging has become a good alternative to the body anatomization technique [76]. IEISAP has been applied to endow the CT contrast agents with tumor-targeting ability [77,78,79] and long tumor retention effect [78,79], such as gold nanoparticles (AuNPs) [77,79] and ^89^Zr [78]. For instance, Sun et al. reported a tumor-specific AuNP-based nanoprobe based on IEISAP for dual CT/optical imaging of cancer [77]. AuNPs were first linked with glycol chitosan (GC) polymers to produce physiologically stable and tumor-targeting GC-AuNPs. GC-AuNPs were further chemically conjugated with an MMP-responsive fluorescent unit (Cy5.5-Gly-Pro-Leu-Gly-Val-Arg-Gly-Lys(BHQ)-Gly-Gly), which was obtained by coupling the NIR fluorescent dye (Cy5.5) and black hole quencher (BHQ) to both ends of the MMP cleavable peptide, resulting in MMP-GC-AuNPs. In MMP-GC-AuNPs, the fluorescence of Cy5.5 was strongly quenched, due to the combinational quenching effect from the gold particle surface and the organic BHQ. The quenched fluorescence of Cy5.5 was recovered when the peptides were cleaved by MMPs that were overexpressed in the tumor tissue. The authors demonstrated that MMP-GC-AuNPs could effectively accumulate in the tumor tissue, visualizing the tumor tissue using CT with great spatial resolution and optical imaging with excellent sensitivity simultaneously in the tumor-bearing mouse model, which offered not only accurate tumor anatomical information, but also MMP-dependent biological information.

### 2.6. Dual/Multimodal Imaging

Dual/multimodal imaging technologies consist of at least two imaging functions combined in one imaging probe, and they are required for advancing biomedical and clinical research. Ideal dual/multimodal imaging approaches should be synergistically combined to overcome the weak points of each imaging method, offering distinct imaging information. Controlling the self-assembly process to simultaneously activate dual/multimodal imaging signals in a small-molecule probe is challenging. IEISAP has been successfully deployed to develop activatable dual/multimodal probes for in vivo imaging of tumor and enzyme activity in real time, including dual CT/optical imaging [77], multimodal fluorescence/PET/CT imaging [78], and dual NIR fluorescence/MR imaging [80]. For instance, Yan et al. rationally designed and fabricated an activatable bimodal probe (P-CyFF-Gd) for molecular imaging using NIR fluorescence and MR by combining a fluorogenic reaction with enzyme-responsive in situ self-assembly [80]. P-CyFF-Gd was composed of a prequenched NIR fluorophore (merocyanine, Cy-Cl), linked with a phosphate group (−PO_3_H) that could be recognized by ALP, a paramagnetic DOTA-Gd chelate for MRI, and a hydrophobic dipeptide Phe-Phe (FF) linker to promote self-assembly (Figure 4a). P-CyFF-Gd was water-soluble due to the presence of the hydrophilic −PO_3_H and DOTA-Gd ligands, and it displayed quenched NIR fluorescence and low *r*_1_ relaxivity. Upon systemic administration, P-CyFF-Gd could easily extravasate and deeply diffuse into tumor tissues because of its hydrophilicity and small molecular size (Figure 4b). In tumor tissues that contain ALP, ALP dephosphorylated P-CyFF-Gd to produce hydrophobic CyFF-Gd, emitting NIR fluorescence at 710 nm. Meanwhile, the FF dipeptide of CyFF-Gd provided effective intermolecular interactions to induce molecular self-assembly, resulting in the formation of fluorescent and magnetic NPs. The assembled NPs possessed a significantly larger molecular size than P-CyFF-Gd, which likely reduced molecular rotation and increased tumbling time (*τ*_R_) of Gd-chelates, thus increasing *r*_1_ relaxivity. Furthermore, the NPs were also prone to bind to the plasma membrane, which could facilitate their cellular uptake and lysosomal localization via endocytosis. The prolonged retention of the NPs in ALP-expressing tumors could be realized, whereas residual P-CyFF-Gd was likely washed away from ALP-negative normal tissues to avoid side effects. Therefore, the formation of NPs simultaneously achieved the enhancements of the NIR fluorescence at 710 nm (>70 folds) and the *r*_1_ relaxivity (~2.3-fold), allowing the real-time detection of ALP activity in live tumor cells and mice with high sensitivity and high spatial resolution. Similar strategies can be adopted to construct other enzyme-activatable bimodal sensors for in-situ and real-time tracking of enzyme activity and location.

In this section, we summarized the achievements regarding the use of IEISAP materials for tumor imaging, enzyme activity assay, and bacterial detection through various imaging modalities, including fluorescence imaging, PA imaging, MRI, PET imaging, CT imaging, and dual/multimodal imaging. These materials are formed through in situ self-assembly of the peptide building blocks, and are then activated by different enzymes in target areas. Therefore, imaging moieties loaded in the IEISAP materials can accumulate in the target regions with enhanced accessibility and retention, enabling specific and sensitive tumor imaging, enzyme activity assay, and bacterial detection both in vitro and in vivo.

## 3. Disease Treatments

In addition to its wide applications for imaging diseases and molecules, IEISAP has also found diverse uses, including cancer therapy, antibacterial treatment, antiinflammation application, and others, which are summarized below. Many diseases are accompanied by the overexpression of kinases and phosphatases, enabling the utilization of these enzymes in peptide assembly in living cells to construct therapeutic agents for disease treatments [38]. Compared with external stimuli, IEISAP might be more suitable in the utilization of biological environments for disease treatments.

### 3.1. Cancer Therapy

The development of functional biomaterials with anticancer activities represents an important research direction for combating cancers. In particular, developing materials capable of targeting the hallmarks of cancer is crucial to realize targeted cancer treatment. Among the various cancer hallmarks, some enzymes that are overexpressed by cancer cells have attracted the interest of many researchers [81]. These cancer cell-overexpressed enzymes enable the IEISAP materials to be selectively formed in cancer cells rather than in normal cells [82], and accordingly, IEISAP has been exploited to develop advanced cancer therapies. Generally, IEISAP materials have been implemented for (1) killing cancer cells directly, (2) delivering anticancer drugs, (3) killing cancer cells in combination with conventional anticancer drugs, and (4) cancer theranostics.

#### 3.1.1. IEISAP Materials That Kill Cancer Cells Directly

Unintended and unregulated IEISAP can be toxic to mammalian cells, which can be adapted for disease treatment by making it happen at the pathological locations. Generally, for the direct application of IEISAP materials for cancer treatment, an enzyme changes the molecular structure of a peptide precursor from a soluble hydrophilic state to a self-assembling unit to form an ordered structure within the cell or pericellular space. It is possible to use peptide-based small molecules to intracellularly polymerize and self-organize into three-dimensional (3D) nanostructures, which can be utilized as drug-free agents for cancer therapy. Therefore, drug-free IEISAP materials can be used directly as nanomedicines to modulate the fate of cancer cells [83,84,85,86,87,88,89,90,91,92,93,94,95,96,97,98,99,100,101,102,103], leading to tumor apoptosis [83,84,85,86,87,88,89,90,91,92,93,94,95,96,102], necroptosis [93,94,95,97,102], autophagy [63], or cellular stress [98,99] (Table 1). Most IEISAP materials cause apoptosis of cancer cells in diverse manners, such as blocking cellular mass exchange [83], disrupting mitochondria to release cytochrome c [84], inducing endoplasmic reticulum stress by ROS [90], and selectively degrading programmed death-ligand 1 (PD-L1) [92] in cancer cells (Table 1).

The most frequently used enzyme that can induce the formation of IEISAP materials in cells or in vivo is ALP [83,84,86,87,88,89,90,91,92,94,95,96,97,100,103], and the less frequently used ones include esterase [99], carboxylesterase (CES) [93,100], transglutaminase (TGase) [85], ATG4B [63], reductase [90], furin [101], and trypsin [102]. Since cancer cells have higher levels of these enzymes than normal cells, the corresponding IEISAP materials may have the capacity to selectively target and kill cancer cells without harming the normal ones. In ALP-overexpressing cancer cells, the peptide-derived small-molecule building blocks are dephosphorylated by ALP and then self-assembled into nanostructures, including nanonets [83], nanofibrils [87,92], nanoparticles [91,95], and hydrogels [103], which can be used for cancer treatments. For instance, Kuang et al. reported the naphthalene-capped tripeptide D-1 (D-Phe-D-Phe-D-Tyr) in which Tyr could be dephosphorylated by ALP to generate the hydrogelator D-2 in the pericellular space, once encountering cancer cells overexpressing ALP, such as HeLa, MES-SA, and MES-SA/Dx5 [83]. The accumulation of D-2 resulted in the formation of a nanofibril network as the scaffold of a hydrogel that encased secretory proteins and blocked cellular uptake, which led to decreased cellular migration and adhesion, and thus caused cancer cell apoptosis. The IEISAP-based anticancer drugs can be integrated into other cancer therapies, such as chemotherapy [63], radiotherapy [96], immunotherapy [88,91,92], and phototherapy [96], to achieve synergistic therapies. Efforts have also been made to promote the formation of the anticancer IEISAP nanomaterials in subcellular organelles, such as mitochondria [84,91], which can realize the spatial control of anticancer drugs within the cancer cells to achieve better therapeutic outcomes.

#### 3.1.2. IEISAP Materials for Delivering Anticancer Drugs

The IEISAP materials have shown great potential for their application in drug delivery, because they can increase therapeutic efficacy and reduce undesired side effects. Generally, conventional anticancer drugs, such as taxol [104,105,106,107,108], paclitaxel [109], 10-hydroxyl camptothecin [110], aldoxorubicin (aldox) [78], cisplatin or its derivative [111,112], and camptothecin [113], photosensitizers, such as AIE luminogen (TPE-Py) [51] and ICG [114], photothermal agents, such as cypate [115] and gold nanoparticles [116], and autophagy inhibitor hydroxychloroquine [113,115], have been loaded to IEISAP materials by chemical conjugation to generate drug-peptide hybrids. Upon encountering various enzymes in vivo, such as ALP [105,106,107,110,111,115], furin [104], MMPs [109,112], and CES [115], these drug-peptide hybrids can be assembled into different nanostructures, mainly nanofibers [78,105,106,107,110,111,112], nanoparticles [104,109,115], and aggregates [51].

We will first introduce the applications of drug-peptide hybrids in which the drugs refer to conventional anticancer drugs. Once these drug-peptide hybrids are inside the cells, the loaded drugs can be released by cleavage of conjugation bonds or disassembly of nanostructures. As an example, Yuan et al. proposed a strategy on the basis of intracellular furin-instructed assembly of a taxol-peptide complex to combat multidrug resistance [104]. The authors rationally designed the taxol-peptide complex Ac-Arg-Val-Arg-Arg-Cys(StBu)-Lys(taxol)-2-cyanobenzothiazole (CBT-Taxol) by introducing the following components: peptide RVRR for furin cleavage and improving the cellular uptake of CBT-Taxol, a disulfide-functionalized cysteine (Cys), a lysine (Lys) whose side chain was linked with taxol, and a 2-cyanobenzothiazole (CBT). Once entering furin-overexpressing cancer cells (i.e., HCT 116 cells in the study), CBT-Taxol went through the reduction by GSH and a condensation reaction induced by furin cleavage, generating hydrophobic oligomers (mainly dimers). The oligomers then self-assembled into taxol nanoparticles (Taxol-NPs) that could tightly bind to the membranous organelles, such as Golgi bodies, due to their large sizes and hydrophobicity. As a result, Taxol-NPs were not easily exported from the cells by P-glycoprotein, exhibiting prolonged retention inside cells. Free taxol was gradually released via cleaving the ester bonds in Taxol-NPs through esterases in cells, and continuously bound to the tubulin to kill multidrug-resistant cancer cells. In comparison with taxol, CBT-Taxol showed a 4.5- and 1.5-fold increase in the anti-multidrug resistance effect on taxol-resistant HCT 116 cancer cells and tumors, respectively, without causing toxicity to the cells or the mice.

Besides delivering conventional anticancer drugs, IEISAP materials have also been successfully leveraged to deliver photoresponsive agents or nanoparticles for cancer treatment. For instance, we rationally designed an ALP and CES dual-enzyme-controlled IEISAP prodrug to enhance the therapeutic efficacy of mild-temperature photothermal therapy (PTT) through autophagy inhibition [115]. The prodrug Cypate-Phe-Phe-Lys(SA-HCQ)-Tyr(H_2_PO_3_)-OH (Cyp-HCQ-Yp) was composed of the following three components: a well-known tetrapeptide Phe-Phe-Lys-Tyr(H_2_PO_3_)-OH for ALP-triggered self-assembly, an NIR cyanine dye cypate (Cyp) for PTT and enhancing the peptide self-assembly, and an autophagy inhibitor hydroxychloroquine (HCQ) that was conjugated with the peptide scaffold through an ester bond which can be cleaved by intratumoral CES (Figure 5a). During the penetration process into cancer cells, Cyp-HCQ-Yp was first dephosphorylated by ALP on the plasma membrane to produce Cyp-HCQ-Y (Figure 5b). Cyp-HCQ-Y was then hydrolyzed by CES in the cytoplasm, generating HCQ and Cyp-Y, the latter of which self-assembled into nanoparticles (Cyp-Y-NP). The released HCQ reduced the resistance of cancer cells to heat by repressing autophagy, making the cells more sensitive to mild hyperthermia, while the generation of Cyp-Y-NP increased the localization and accumulation of cypate within tumor cells, both contributing to the enhanced photothermal effect under NIR light irradiation. In tumor cells, Cyp-HCQ-Yp induced a much higher level of apoptosis/necrosis than Cyp-Yp or HCQ-Yp (the molecular structures of HCQ-Yp and Cyp-Yp are shown in Figure 5c). In in vivo experiments, the administrated Cyp-HCQ-Yp generated a mild temperature (less than 44 °C) in the tumor site with laser irradiation, fully eradicating the tumor without any adverse effect. This “tandem enzymatic self-assembly” strategy represents a potential approach to develop smart prodrugs for synergistic cancer therapy with elevated tumor-targeting, sustained drug release, improved efficacy, and enhanced safety.

In addition to chemical conjugation, certain anticancer compounds can also be delivered by physical encapsulation into IEISAP materials, such as doxorubicin [48] and red phycoerythrin [48]. IEISAP can realize the drug delivery to not only cells, but also subcellular compartments, such as the nucleus [110] and mitochondrion [48]. The application of in situ enzymatic formation of supramolecular nanostructures by drug-peptide conjugates for drug delivery to cancer cells/tissues offers several advantages, including high and controllable drug loading [108,112], sustained or controlled release of drugs in biological environments [104,105,112,115], higher selectivity to their targets [51,111,114], enhanced cellular uptake [104,107,110], and better biocompatibility to normal cells and tissues [104,111].

#### 3.1.3. IEISAP Materials Used in Combination with Traditional Chemotherapeutic Drugs for Cancer Treatment

Except being employed alone as nanomedicines or as drug carriers for anticancer drugs, IEISAP materials have been combined with traditional chemotherapeutic drugs, such as cisplatin [44,117], dichloroacetate [44], paclitaxel (taxol) [44], and nuclear factor-kappa B (NF-κB) inhibitor [97], for synergistic cancer treatments. On one hand, the presence of innocuous IEISAP materials notably elevated the anticancer activities of the chemotherapeutic drugs [44,117]. For instance, Li et al. designed and constructed a small peptide precursor of which the ester bond could be cleaved by CES to produce the peptide that could self-assemble into nanofibers in water (Figure 6a) [117]. With the optimal concentrations, the precursors were not toxic to cells, but they significantly increased the activity of cisplatin toward the drug-resistant ovarian cancer cells, without elevating the systemic burden or side effects. This study proposes a feasible means to combine IEISAP materials with chemotherapeutic drugs. On the other hand, the presence of NF-κB inhibitors can turn the otherwise innocuous IEISAP materials lethal to the cancer cells via necroptosis. Zhou et al. synthesized a C-terminal methylated phosphotetrapeptide (pTP-Me) that could be dephosphorlated by ALP in the cellular milieu of cancer cells, such as Saos-2, to generate TP-Me (Figure 6b) [97]. TP-Me then formed nanofibers and led to the inductive expression of tumor necrosis factor receptor 2 (TNFR2) and reduced expression of three key proteins at the upstream of NF-κB pathway, including PI3K, Akt, and MEKK3, which hardly affected cell viability. Nevertheless, the addition of the inhibitor (BAY 11-7085) that targeted NF-κB further downregulated the expression of the upstream proteins and substantially reduced cancer cell viability by approximately one order of magnitude, without compromising the selectivity toward cancer cells, eventually leading to the death of Saos-2 by necroptosis. This research provides a versatile way to employ crucial regulatory signal pathways as potential therapeutic targets. Overall, these studies indicate that IEISAP represents a promising approach in developing combination therapies for cancer treatment.

#### 3.1.4. IEISAP Materials for Cancer Theranostics

The combination of diagnostic and therapeutic properties within a single formulation is highly desirable for precision cancer treatment, which can benefit early personalized diagnosis and subsequent specific therapy to maximize therapeutic efficacy with minimal side effects by imaging the focus location, monitoring drug delivery/release, realizing targeted treatment, or even evaluating therapeutic effect. Recently, some theranostic systems have been developed on the basis of IEISAP materials by conjugating peptides with fluorescent dyes, such as AIEgen [45,51,53,54,118], cyanine dyes [40], and 1,8-naphthalic anhydride [58], PA agents [62,63], MRI agents [69], and CT contrast agents [79].

For example, Yuan et al. constructed a chemotherapeutic Pt(IV) prodrug, whose two axial positions were modified with c-RGD tripeptide for targeting cancer cells overexpressing integrin α_v_β_3_, and a caspase-3-specific DEVD peptide that was linked with a tetraphenylsilole (TPS) fluorophore with aggregation-induced emission (AIE) characteristic for sensing apoptosis, respectively (Figure 7) [53]. The prodrug could specifically bind to the U87-MG cancer cells overexpressing integrin α_v_β_3_ to assist cellular internalization. Once inside U87-MG cells, the Pt(IV) prodrug could be reduced to the active Pt(II) drug and simultaneously release the apoptosis sensor TPS-DEVD that was nonemissive in aqueous media because of free rotation of the phenylene rings. The reduced Pt(II) drug could cause cell apoptosis and activate caspase-3 enzyme that cleaved DEVD to generate hydrophobic TPS. TPS was prone to aggregate, leading to the restricted rotation of the phenyl rings that could turn on the fluorescence. The intensity of the apoptosis-triggered fluorescence displayed good correlation with the prodrug concentration and the cell viability. Therefore, the authors realized the real-time and in-situ imaging of apoptosis caused by the Pt(II) drug, which was utilized for early evaluation of the therapeutic responses of the Pt(IV) prodrug. Clearly, this theranostic system with a built-in apoptosis probe allows the drug delivery to cancer cells with high selectivity and early evaluation of the drug therapeutic effect, guiding therapeutic decisions.

### 3.2. Antibacterial Treatment

In contrast to the extensive studies on applying IEISAP for combating cancers, only a few examples that adopt IEISAP for killing bacteria have been reported [28,29,30,119]. In these studies, it has been demonstrated that enzyme-triggered assembly of peptide derivatives using ALP [28,29,30] or gelatinase [119] can induce bacterial cell death by the intracellular formation of nanofibers. For instance, Qi et al. designed and synthesized a chitosan–peptide conjugate (CPC), whose morphology could be changed by gelatinase for the treatment of bacterial infection [119]. The CPC was composed of a chitosan backbone, an enzyme-cleavable peptide GPLGVRGC with a poly(ethylene glycol) (PEG, molecular weight: 2 kDa) terminal (EPEG), and an antibacterial peptide CGGGKLAKLAKKLAKLAK (KLAK). The CPCs self-assembled into nanoparticles in aqueous solution (Figure 8a), which could be transformed into nanofibers after treatment of gelatinase secreted by a broad spectrum of bacterial species. Upon arriving at an infected microenvironment containing geletinase, the EPEG segments were cleaved by gelatinase; thus, the protective PEG layer of the CPC nanoparticles was peeled off, leading to the collapse of the hydrophobic–hydrophilic balance of the nanoparticles, and thus the reorganization of them into fibrous nanostructures through chain–chain interaction of chitosans (Figure 8b). After this structural transformation, the *α*-helical structures of KLAK were exposed, enabling multivalent cooperative electrostatic interactions of CPC nanofibers with bacteria to damage cell membranes. This enzyme-triggered morphological transformation endowed CPCs with enhanced binding ability to bacterial cells, as well as increasing accumulation and prolonged retention at the bacteria-infected sites, resulting in improved antibacterial performance. This on-site transformation of antibacterial nanoparticles with the assistance of enzymes offers a new idea for designing robust antibacterial materials.

### 3.3. Antiinflammation Application

Inflammation is a reflexive response of an organism to injury, infection, mechanical irritation, or the binding of antibodies to antigens. Although circumscribed inflammation is beneficial for human health, excessive or persistent inflammation causes allergies, autoimmune diseases, asthma, and sepsis, which are the major causes of illness and death. IEISAP has been implemented to improve the efficacy of a traditional antiinflammatory drug dexamethasone (DEX) by increasing its cellular uptake and retention and prolonging drug release [120], or decreasing its ROS-caused side effects [121]. For example, Song et al. reported the self-amplifying assembly of peptides in response to the expression of the enzyme NAD(P)H quinone dehydrogenase 1 (NQO1) in inflamed macrophages for improving the efficacy of DEX (Figure 9) [121]. The authors coassembled a quinone propionic acid (QPA)-linked pentapeptide AmpFQ and its derivative (AmpFQ−ETGE), containing an ETGE sequence from the nuclear factor erythroid 2-related factor 2 (Nrf2) domain, leading to the formation of AmpFQB nanoparticles. The AmpFQB nanoparticles were further loaded with DEX, forming DEX@AmpFQ nanoparticles. In the presence of NQO1, QPA was reduced and released from the peptides AmpFQ and AmpFQ−ETGE to generate cyclic hydroquinone propionic acid (chQPA), releasing DEX to downregulate proinflammatory cytokines, and the peptide FF-Amp-FF (AmpF) to assemble into nanofibrils (AmpFB) under physiological conditions. ETGE could bind to Kelch epichlorohydrin (ECH)-associated protein 1 (Keap1), which dissociated the Nrf2−Keap1 complex to release and activate Nrf2. The activated Nrf2 could increase the expression of NQO1, which subsequently elevated the peptide assembly into AmpFB, establishing a close amplifying relationship between the NQO1 level and the peptide assembly in the macrophages. Therefore, the coassembly of AmpFQ with AmpFQ−ETGE gave rise to the NQO1-amplifying assembling system AmpFQB. Meanwhile, AmpFQB could passively target acutely injured lungs through the enhanced permeability and retention effect. DEX@AmpFQ was utilized for improved antiinflammatory treatment of acute lung injury by simultaneous downregulation of proinflammatory cytokines and alleviation of the ROS-induced side effect. This study presents a strategy to realize the self-amplifying peptide assembly by associating the enzyme expression with the assembly process, overcoming the limitation that IEISAP strongly depends on the expression levels of naturally occurring enzymes.

### 3.4. Others

In addition to the above-mentioned anticancer, antibacterial, and antiinflammation applications, IEISAP can also be applied in the delivery of ocular [31] and cardiovascular [122] drugs. Hu et al. implemented the IEISAP strategy for ocular drug delivery [31]. The nonsteroidal antiinflammatory drug ibuprofen (IBF) was covalently linked with a self-assembling peptide GFFpY via a hydrolyzable ester bond to generate a phosphorylated peptide-drug precursor (IBF-HYD-GFFpY). The dephosphorylation of IBF-HYD-GFFpY was caused by the catalysis of ALP in the tear fluid, yielding IBF-HYD-GFFY that subsequently self-assembled into nanofibers to realize the sustained release of IBF over 96 h through the hydrolytic cleavage of the ester bond by esterase. The resulting IBF-HYD-GFFpY eye drops dramatically improved precorneal retention of drugs without causing eye irritation. In a rabbit model of endotoxin-induced uveitis, the 0.5 wt% IBF-HYD-GFFpY eye drops could reduce the influx of macrophages and leukocytes and exhibit therapeutic efficacy comparable with that of the 0.1 wt% diclofenac eye drops used in clinic. Collectively, this study provides an approach to exploit IEISAP materials for effectively delivering ocular drugs with targeted and prolonged retention at the lesion area.

In another study, Nguyen et al. presented an approach based on IEISAP for targeted accumulation and prolonged retention in heart tissue after myocardial infarction (MI), using a brush peptide–polymer amphiphile (PPA) [122]. The PPA was composed of a polynorbornene backbone with peptide sequences specific for the recognition of MMP-2 and MMP-9, the two enzymes upregulated in the acute MI, which formed nanoparticles in vitro. After being administered via intravenous injection, the PPA nanoparticles freely circulated in the bloodstream until arriving at the infarct via the leaky post-MI vasculature, where the nanoparticles were transformed into network-like scaffold caused by MMPs, resulting in long retention within the injured area for up to 28 d post-injection. This IEISAP-based morphological transition of PPAs offers a promising approach for delivering drugs post-MI by intravenous injection, thus avoiding the need for risky intramyocardial injections.

## 4. Conclusions

In this review, we present an overview on the advances in using IEISAP materials for different biomedical applications, such as bioimaging (fluorescence, PA, MR, PET, CT, and dual/multimodal imaging) and disease treatment (anticancer, antibacterial, antiinflammation, and other applications), demonstrating that IEISAP represents a powerful method to construct robust strategies for disease diagnosis and treatment, particularly for cancer treatment. The most significant advantage of IEISAP is that it can enable diagnostic and therapeutic agents to achieve targeted and increased accumulation at the lesion sites with prolonged retention, leading to improved performance and reduced side effects. Additionally, IEISAP offers other benefits, such as controllable cellular fates, long circulation time, and satisfactory biocompatibility. Nonetheless, some challenges still remain to be overcome in this field.

First, fabrication of IEISAP materials for biomedical applications in a controllable and predictable manner has not been completely realized yet, since elaborate manipulation of the self-assembly process of peptides in cellular environments and in tissues is still a challenge in supramolecular chemistry. In the future, more synthetic methods, conjugation approaches, and self-assembly strategies should be developed for this purpose. Second, efficient observation techniques that can monitor real-time morphological changes in IEISAP materials and characterize the dynamic behaviors of peptide assemblies in living cells and organisms are still lacking. This is a formidable challenge that needs to be addressed urgently. Third, IEISAP materials may change their structures in sophisticated in vivo environments because of their dynamic and reversible nature, thereby altering their properties and functions. As such, careful analysis of the in vivo structural changes in IEISAP materials and their correlation with the diagnostic and therapeutic performance will facilitate precise and effective disease diagnosis and therapy. Fourth, most of the currently reported applications of IEISAP focus on cancer therapy, and potential uses of IEISAP in other biomedical fields, such as microbial infections (especially for viral infections), cardiovascular diseases, and neurodegenerative diseases, should be checked. We believe that the continuous collaborative efforts among multidisciplinary scientists will bring about more practical applications of IEISAP in different areas of biomedicine in the future. Fifth, despite the advantageous properties of IEISAP materials, their practical applications and clinical translation can be impeded by their relatively poor stability. Perhaps unnatural amino acids or other specifically designed amino acid mimetics can be incorporated outside the enzyme recognition sites in peptide precursors to improve the metabolic stability of IEISAP materials in vivo. Finally, it is essential to strictly evaluate the long-term pharmacokinetics and biosafety of IEISAP materials in vivo, which is important for their successful clinical translation. Collectively, great efforts are still needed for the preparation, characterization, application, and safety evaluation of IEISAP materials to ensure the bright prospect of IEISAP in intelligent nanomedicine.

## Data Availability

Data sharing not applicable.

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
