# Peer review of "Intracellular Enzyme-Instructed Self-Assembly of Peptides (IEISAP) for Biomedical Applications"

_molecules, 2022, doi:10.3390/molecules27196557_

Round 1

Reviewer 1 Report

Lin et al. carefully summarized the progress on the use of enzyme-regulated self-assembly of peptides for diverse biomedical applications. Given that very few reviews concerning the achievements of the intracellular peptide assembly induced by enzymes have been reported, this is a very comprehensive and exhaustive paper. The review should be of interest for researchers working on molecular assembly, biomaterials, and the related topics. Overall, the paper is well-written and well-illustrated, with a good organization of the topic. In my opinion, the manuscript can be published after being paying attention to some minor points as listed below:

(1)   Please adjust the inverted text direction and unify the word size in Scheme 1 so as to be more convenient to read.

(2)   Please unify the meaning of “IEISAP”. IEISAP was an abbreviation of intracellular enzyme-instructed self-assembly of peptides on page 1, line 8, while it represented in situ enzyme-instructed self-assembly of peptides on page 2, line 73.

(3)   For section “3. Disease treatments”, I suggest the authors to add 1-2 additional sentences to introduce the related background.

(4)   For Figure 6, please rearrange this figure to make it clearer.

(5)   Please check the expression of the “IEISAP materials” in section 3.1.1, since in sections 3.1.2, 3.1.3, and 3.1.4, the authors wrote “IEISAP nanomaterials”.

Author Response

General Comment: Lin et al. carefully summarized the progress on the use of enzyme-regulated self-assembly of peptides for diverse biomedical applications. Given that very few reviews concerning the achievements of the intracellular peptide assembly induced by enzymes have been reported, this is a very comprehensive and exhaustive paper. The review should be of interest for researchers working on molecular assembly, biomaterials, and the related topics. Overall, the paper is well-written and well-illustrated, with a good organization of the topic. In my opinion, the manuscript can be published after being paying attention to some minor points as listed below:

Response: We are extremely grateful for the precious time, invaluable expertise, and superb professionalism the respected reviewer has put in improving the quality of our paper! 

Comment 1: Please adjust the inverted text direction and unify the word size in Scheme 1 so as to be more convenient to read.

Response: Revised as suggested. Thanks a lot!

Comment 2: Please unify the meaning of “IEISAP”. IEISAP was an abbreviation of intracellular enzyme-instructed self-assembly of peptides on page 1, line 8, while it represented in situ enzyme-instructed self-assembly of peptides on page 2, line 73.

Response: As suggested, we have changed “in situ enzyme-instructed self-assembly of peptides” on page 2, line 73 to “intracellular enzyme-instructed self-assembly of peptides” to unify the meaning of “IEISAP”.

Comment 3: For section “3. Disease treatments”, I suggest the authors to add 1-2 additional sentences to introduce the related background.

Response: As suggested, the additional sentences have been added to introduce the related background: “Many diseases are accompanied by the overexpression of kinases and phosphatases, enabling the utilization of these enzymes in peptide assembly in living cells to construct therapeutic agents for disease treatments [38]. Compared with external stimuli, IEISAP might be more suitable for utilizing the biological environments for disease treatments.”

Comment4: For Figure 6, please rearrange this figure to make it clearer.

Response: We have rearranged Figure 6 to make it clearer.

Comment 5: Please check the expression of the “IEISAP materials” in section 3.1.1, since in sections 3.1.2, 3.1.3, and 3.1.4, the authors wrote “IEISAP nanomaterials”.

Response: We have changed “IEISAP nanomaterials” to “IEISAP materials” which we think is more accurate, given that in some cases the structure of IEISAP is not in the nanoscale such as hydrogel.

Author Response

General Comment: This article provides a comprehensive overview of the progress in the biomedical applications of intracellular enzyme-instructed self-assembly of peptides (IEISAP), including bioimaging like fluorescence imaging, photoacoustic imaging, magnetic resonance imaging, positron-emission tomography imaging, radiation imaging, and multimodal imaging, and disease treatment such as anticancer, antibacterial, and antiinflammation therapies. The authors made a careful and critical examination of the topic. The review is well-written, and I believe it may attract much interest from the readers in the related field. I would like to recommend the publication of the paper after minor revision.

Response: We are extremely grateful for the precious time, invaluable expertise, and superb professionalism the respected reviewer has put in improving the quality of our paper. We want to thank the reviewer for stating that “The authors made a careful and critical examination of the topic. The review is well-written, and I believe it may attract much interest from the readers in the related field. I would like to recommend the publication of the paper after minor revision.”.

Comment 1: Did the study mention in Line 102-103 really performed in vivo or intracellularly, please double check to make sure the description is correct.

Response: We have carefully checked the related paper to make sure the study was performed in vivo.

Comment 2: In section 3.1.3., the authors are suggested to add more discussions for the example they introduced.

Response: As suggested, more discussion was added on the examples we introduced as below.

“This study demonstrate an easy yet fundamental and novel means to integrate IEISAP materials into combination therapy using cisplatin with no systemic burden or side effects.”

“This research provides a versatile way to employ crucial regulatory signal pathways into potential therapeutic targets.”

Comment 3: The paper is well written, but several typos should be corrected. For example, “Societ” should be “Society” in Line 538, “dia gnosis” should be “diagnosis” in Line 668, “therebyaltering” should be “thereby altering” in Line 685.

Response: The typos mentioned by the reviewer have been corrected in our revised manuscript. Also, We have looked through the whole manuscript and tried our best to correct the typo errors we found.

Comment 4:  “as we as review below” in section 2 should be “as we will review below”.

Response: Revised as suggested.

Comment 5: The abbreviated journal name in the Reference should be double-checked. For example, “Angew. Chem., Int. Ed.” should be “Angew. Chem. Int. Ed.”, and “J. Controlled Release” should be “J. Control. Release”

Response: The mistakes of the abbreviated journal names in the Reference mentioned by the reviewer have been corrected in our revised manuscript. Also, We have double-checked the abbreviated journal names in the Reference.

Reviewer 3 Report

In this manuscript, authors gave a review for the advances in the applications of IEISAP, including various bioimaging techniques in cancer tissues and cells, bacteria, and enzyme activity.

This review is comprehensive and timely.

Thus I enthusiastically support acceptance.

Author Response

General Comment: In this manuscript, authors gave a review for the advances in the applications of IEISAP, including various bioimaging techniques in cancer tissues and cells, bacteria, and enzyme activity.

This review is comprehensive and timely.

Thus I enthusiastically support acceptance.

Response: We are extremely grateful for the precious time, invaluable expertise, and superb professionalism the respected reviewer has put in improving the quality of our paper. We want to thank the reviewer for stating that “This review is comprehensive and timely. Thus I enthusiastically support acceptance.”.

Reviewer 4 Report

The manuscript by Wu et al. reviews about the intracellular enzyme-instructed self-assembly of peptides(IEISAP) for biomedical applications including imaging applications and disease treatment. This manuscript provides a brief background with its novelty. The authors well-categorized the research in terms of IEISAP, based on the applications. In additions, the contents are dens and well-written thus making it easy to read and understand. Therefore, I recommend this manuscript to be accepted, after the authors consider some minor points which could improve the potential impact of this manuscript.

1.     Page 3, Line 109-113: The authors assert that the review paper in terms of IEISAP for biomedical applications is still lacking. As my knowledge, there are some review papers for the biomedical applications of IESAP. Ex) Enzymatic Noncovalent Synthesis of Supramolecular Soft Matter for Biomedical Applications. Matter 2019, 1, 1127-1147. The authors maybe want to introduce the review papers including the similar contents.

2.     Page 11, Line 406-409 and Page 12, Table 1: Kindly, the authors summarize the IESAP materials used as anticancer drugs directly and the enzymes play the role as the self-assembly trigger. There are some research papers missed in this manuscript. Ex) Furin-mediated intracellular self-assembly of olsalazine nanoparticles for enhanced magnetic resonance imaging and tumour therapy. Nat. Mat. 2019, 18, 1376-1383; Trypsin-Instructed Self-Assembly on Endoplasmic Reticulum for Selectively Inhibiting Cancer Cells. Adv. Healthc. Mater. 2021, 10, 2000416. Richer contents will be of great help to readers.

3.     There are some typos.

A.     Line 414 : nanopariticle
B.     Line 423  : Immuotherapy

C.     Line 685  : therebyaltering

Author Response

General Comment: The manuscript by Wu et al. reviews about the intracellular enzyme-instructed self-assembly of peptides(IEISAP) for biomedical applications including imaging applications and disease treatment. This manuscript provides a brief background with its novelty. The authors well-categorized the research in terms of IEISAP, based on the applications. In additions, the contents are dens and well-written thus making it easy to read and understand. Therefore, I recommend this manuscript to be accepted, after the authors consider some minor points which could improve the potential impact of this manuscript.

Response: We are extremely grateful for the precious time, invaluable expertise, and superb professionalism the respected reviewer has put in improving the quality of our paper. We want to thank the reviewer for stating that “This manuscript provides a brief background with its novelty. The authors well-categorized the research in terms of IEISAP, based on the applications. In additions, the contents are dens and well-written thus making it easy to read and understand. Therefore, I recommend this manuscript to be accepted, after the authors consider some minor points which could improve the potential impact of this manuscript.”.

Comment 1:  Page 3, Line 109-113: The authors assert that the review paper in terms of IEISAP for biomedical applications is still lacking. As my knowledge, there are some review papers for the biomedical applications of IESAP. Ex) Enzymatic Noncovalent Synthesis of Supramolecular Soft Matter for Biomedical Applications. Matter 2019, 1, 1127-1147. The authors maybe want to introduce the review papers including the similar contents.

Response: We thanks the reviewer for recommending the related excellent review papers which has been cited in the revised manuscript. The recommended paper covered enzymatic peptide self assembly under both cell-free conditions and cellular conditions, different from our review paper focusing specifically on intracellular enzymatic peptide self assembly.

Comment 2: Page 11, Line 406-409 and Page 12, Table 1: Kindly, the authors summarize the IESAP materials used as anticancer drugs directly and the enzymes play the role as the self-assembly trigger. There are some research papers missed in this manuscript. Ex) Furin-mediated intracellular self-assembly of olsalazine nanoparticles for enhanced magnetic resonance imaging and tumour therapy. Nat. Mat. 2019, 18, 1376-1383; Trypsin-Instructed Self-Assembly on Endoplasmic Reticulum for Selectively Inhibiting Cancer Cells. Adv. Healthc. Mater. 2021, 10, 2000416. Richer contents will be of great help to readers.

Response: We thanks the reviewer for recommending the related excellent papers which have been added in Table 1 in the revised manuscript.

Comment 3: There are some typos.

  1.     Line 414 : nanopariticle
  2.     Line 423  : Immuotherapy
  3.     Line 685  : therebyaltering

Response: The typos mentioned by the reviewer have been corrected in our revised manuscript. Also, We have looked through the whole manuscript and tried our best to correct the typo errors we found.